# Academic Teaching Quality Framework and Performance Evaluation Using Machine Learning

Ahmad Almufarreh , Khaled Mohammed Noaman and Muhammad Noman Saeed *

Deanship of eLearning and Information Technology, Jazan University, Jazan 45142, Saudi Arabia
* Correspondence: msaeed@jazanu.edu.sa

**Abstract:** Higher education institutions' principal goal is to give their learners a high-quality education. The volume of research data gathered in the higher education industry has increased dramatically in recent years due to the fast development of information technologies. The Learning Management System (LMS) also appeared and is bringing courses online for an e-learning model at almost every level of education. Therefore, to ensure the highest level of excellence in the higher education system, finding information for predictions or forecasts about student performance is one of many tasks for ensuring the quality of education. Quality is vital in e-learning for several reasons: content, user experience, credibility, and effectiveness. Overall, quality is essential in e-learning because it helps ensure that learners receive a high-quality education and can effectively apply their knowledge. E-learning systems can be made more effective with machine learning, benefiting all stakeholders of the learning environment. Teachers must be of the highest caliber to get the most out of students and help them graduate as academically competent and well-rounded young adults. This research paper presents a Quality Teaching and Evaluation Framework (QTEF) to ensure teachers' performance, especially in e-learning/distance learning courses. Teacher performance evaluation aims to support educators' professional growth and better student learning environments. Therefore, to maintain the quality level, the QTEF presented in this research is further validated using a machine learning model that predicts the teachers' competence. The results demonstrate that when combined with other factors particularly technical evaluation criteria, as opposed to strongly associated QTEF components, the anticipated result is more accurate. The integration and validation of this framework as well as research on student performance will be performed in the future.

**Keywords:** e-learning; machine learning; teacher performance; learning analytics

## 1. Introduction

Education has been increasingly crucial to the nation's development in recent years due to technological and scientific advancements and the continuous evolution of society. The primary goal is to raise the number of top-notch students for the nation and society [1]. The standardized process of evaluating and measuring an academic's competency as a teacher is termed teacher assessment. A system that encourages teacher learning will be different from one whose goal is to assess teacher proficiency. Such a system of evaluation created with effective teaching in mind recognizes and rewards the improvement of teachers [2].

Teacher effectiveness is an essential factor in student achievement, and research has consistently shown that effective teachers are a crucial ingredient for student success. However, identifying and retaining effective teachers can be challenging for schools and districts. Traditional methods of evaluating teacher performance, such as observations and evaluations by administrators, are subjective and may not accurately reflect a teacher's impact on student learning. On the other hand, machine learning algorithms are objective. They can analyze a large amount of data to identify patterns and correlations that may not be apparent to the human eye.

Assessment of instructors in higher education is crucial to ensuring that students receive a high-quality education and that teachers can deliver the most excellent possible learning experience for their pupils. Educators are learning that a well-designed assessment system may successfully combine professional development with quality assurance in teacher evaluation, rather than encumbering administrators. The assessment framework, therefore, required adaptable assessment methods and additionally offers adaptive learning material selection, embedding, and presentation based on the student's performance [3]. The assessment also helps ensure that teachers meet the institution's expectations and standards. Differentiated systems, yearlong cycles, and active instructor engagements via portfolios, professional discussions, and student achievement proof are some of the new participatory evaluation approaches [4].

Higher education institutions have specific standards and expectations for their faculty, and assessment helps ensure that these standards are being met besides maintaining the quality and reputation of the institution, as well as ensuring that students are receiving a high-quality education. For example, suppose a teacher must meet the institution's expectations regarding teaching practices or interactions with students. In that case, they may be required to improve their performance or may even be at risk of losing their position. It is challenging for the teachers in charge to grasp the learning scenarios of each quality of teaching and provide timely advice and assistance on teaching quality due to the reform and growth of education. Universities have inevitably increased their enrollment [5]. Teaching quality results have always been one of the significant indicators for schools and organizations to analyze the teaching environment and overall quality of teaching in an organization. When conducting teaching evaluation activities, we first need to establish the teaching evaluation indicators. With clear indications, trainers have a basis for comparing and referencing in the classroom instructional process, indicating that teaching evaluation has a directing function.

Assessment is most important for giving teachers insightful feedback on their work and areas for development. It can assist teachers in determining their areas of strength and weakness and in creating plans for dealing with any difficulties they may encounter. Assessment can assist teachers in meeting the requirements of their students more effectively and helping them to improve their teaching methods continuously. A teacher might decide to include more interactive activities or use more multimedia materials, for instance, if students complain that their lectures are not attractive enough. Another critical reason for assessment in higher education is that it helps ensure students receive a well-rounded education. Higher education institutions often have various programs and courses, and students must be exposed to various teaching styles and approaches. Assessment can ensure that students are exposed to diverse teaching practices and receive a well-rounded education.

Agencies that accredit institutes of higher learning in different nations deal with quality supervision and accountability. In addition to marketing their online programs to a large audience, keeping a high standard in their courses, and demonstrating learning outcomes, e-Learning leaders should indeed demonstrate the efficiency and quality of their online courses to these accrediting agencies [6]. In addition to this, the mechanisms for quality assurance and improvement are included in quality frameworks to provide thorough coverage of the variables influencing students' learning experiences. The success and efficiency of the program can be increased by directing these toward an e-learning design framework, which can be advantageous to all individuals involved in the e-learning system and courses, such as administrators, students, and teachers [7]. The key domain features of frameworks presented in Table 1 offer adaptable benchmarks and principles for online learning, as well as techniques that e-learning professionals can utilize. They are responsible to provide a comprehensive view of all the aspects that must be considered when implementing e-learning, based on the stage of the process of implementing online learning or the necessity for quality evaluation at various levels in the organizational chain.

**Table 1.** Quality benchmarks—frameworks for e-learning.

| Benchmarks/Framework | Parameters | Key Components Example |
|---|---|---|
| Council For Higher Education Accreditation [8] | 7 | Institutional Goals<br>Structure Of The Organization<br>Resources For Institutions<br>Instructional Materials<br>Faculty And Learner Support |
| European Association Of Distance Teaching Universities (EADTU) [9] | 6 | Management<br>Curriculum & Course Design<br>Course Delivery<br>Staff And Student Support |
| Quality On The Line: Benchmarks For Success In Internet-Based Distance Education [10] | 24 | Instructional Support<br>Course Structure And Development<br>Student & Faculty Support<br>Evaluation & Assessment Benchmarks |
| Australasian Council on Open, Distance and e-Learning (ACODE) [11] | 8 | Information Technology Systems<br>Services & Support<br>Staff Professional Development &<br>Student Training Development<br>Staff And Student Support |
| National Association For Distance Education And Open Learning In South Africa [12] | 13 | Course, Curriculum Design<br>And Development<br>Evaluation And Assessment |
| Online Learning Consortium Quality Scorecard [13] | 9 | Institutional & Technology Support<br>Instructional Design<br>Course Structure<br>Social & Student Engagement<br>Faculty & Student Support<br>Evaluations & Assessment |
| Blackboard Exemplary Rubric [14] | 17 | Course Design<br>Interaction & Collaboration<br>Assessment<br>Learner Support |
| Quality Matters—QM [15] | 8 | Course Overview And Introduction,<br>Educational Goals<br>Evaluation & Measurement<br>Instructional Content.<br>Learner Interaction & Course Activities<br>Course Technology,<br>Learner Support<br>Usability & Accessibility |
| iNACOL [16] | 11 | Course Design & Management.<br>Online Course Content<br>Instructional Design<br>Technology For Distance Education<br>Assessment For Student |
| Asian Association Of Open Universities (AAOU) [17] | 10 | Policy, Planning & Management.<br>Students & Faculty Profiles<br>Technology Infrastructure,<br>Learning Contents & Media<br>Assessment & Evaluation<br>Research & Social Services<br>Learner Support<br>Design & Curriculum/<br>Course Development |

To summarize, there are compelling reasons to believe that well-designed teacher-evaluation programs will have a direct and long-term impact on individual teacher performance [18]. Overall, the assessment of teachers in higher education is a crucial part of ensuring that students receive a high-quality education and that teachers can provide their students with the best possible learning experience. It provides valuable feedback to teachers, helps ensure that they are meeting the institution's expectations and standards, and helps identify and address any issues or concerns that may be impacting the teacher's performance or the student's learning experience. By prioritizing assessment, higher education institutions can ensure that they provide their students with the best possible education.

## 2. Electronic Learning and Quality in Teaching

### 2.1. Electronic Learning

Electronic learning, or e-learning, is a type of education that involves the use of electronic devices and technologies, such as computers, laptops, tablets, and smartphones, to access educational materials and resources, communicate with teachers and peers, and complete assignments and assessments [19]. E-learning can be used to supplement traditional classroom-based instruction or be the primary mode of learning for students participating in distance education or online degree programs. In addition, it provides students with a virtual learning environment where they can access course materials, submit assignments, participate in discussions, and take assessments. E-learning typically involves learning management systems and software platforms that enable teachers to create and manage online course content.

### 2.2. Quality in Teaching

Ensuring the quality of teaching in distance education is crucial for student success and satisfaction. Several factors can impact the quality of teaching in an electronic learning environment. The use of technology, the effectiveness of the course design, and the level of interaction between students and instructors are some core factors. Faculty members significantly shape the quality of e-learning [20]. Providing ongoing support and professional development for online instructors can also improve the quality of teaching in this environment.

Course design is the main topic for maintaining quality in teaching and learning, along with the technical and assessment part of the course. Various factors influence a quality learning experience for students; however, the institute still places the most significant emphasis on course design. For instance, a teacher might have incorporated several discussion platforms for engagement in the course as part of the planning process. According to students, online learning was more convenient than in-person instruction because it was more adaptable [21].

One factor that can impact the quality of teaching in distance education is the use of technology. Online courses often rely on technology to facilitate communication and deliver course materials, and the effectiveness of this technology can significantly influence the quality of the learning experience. For example, if the technology is reliable and easy to use, it can help student learning and engagement. Another factor that can affect the quality of teaching in electronic learning is the effectiveness of the course design. A poorly designed course can confuse and frustrate students and hinder their progress. A well-designed online course should be structured logically and intuitively, with clear learning objectives, meaningful assessments, and student interaction and feedback opportunities.

Finally, the level of interaction between students and instructors is also essential in determining the quality of teaching in distance education. Some online courses offer little or no direct interaction with instructors, while others may provide regular opportunities for virtual office hours or live lectures. The traits and behaviors of the instructors, as well as the methods and media adopted to give online training, impact learning online. The university's prime objective is to enhance existing methods and offer a reliable method

of delivering education. The provision of high-quality education was made possible by providing the faculty with the necessary knowledge, skills, and capacities [22].

## 3. Quality Teaching and Evaluation Framework—QTEF

Frameworks for teaching and learning offer contextualized, varied techniques that guide students in creating knowledge structures that are precisely and meaningfully ordered while advising them on when and how to apply the skills and knowledge they acquire [23]. Creating engaging and inclusive learning environments, integrating assessment into learning, and aligning learning objectives with classroom activities are all made possible by teaching and learning frameworks, which are research-based models for course design.

To serve as conceptual maps for organizing or revising any course, curriculum, or lesson, frameworks such as Backward Design, which Wiggins and McTighe first introduced in their book "Understanding by Design", may be easily blended and adjusted [24]. Some similar indications were discovered and mentioned after assessing the presented frameworks in Table 1 and standards to offer e-learning courses, as well as guidance on where to concentrate the focus when trying to improve the quality of online learning. These metrics place emphasis on content design, e-learning program management, technology use, faculty and student facilitation, etc., as depicted in Figure 1.

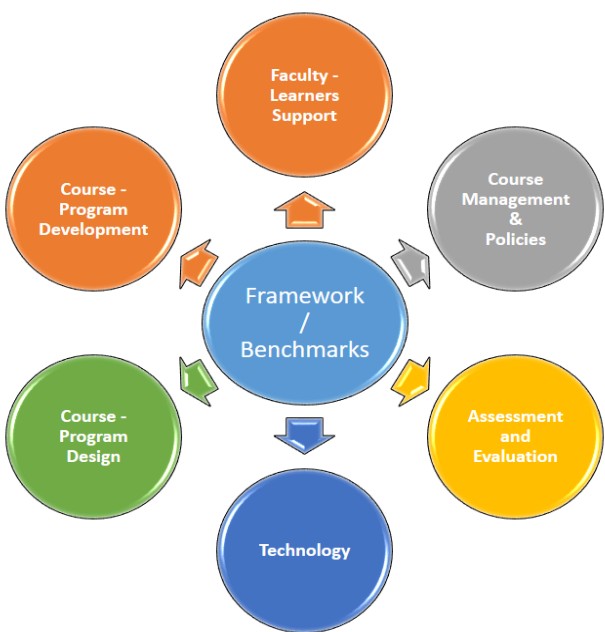

**Figure 1.** Quality indicators for digital education.

The frameworks/benchmarks shown in Table 1 have been developed by leading e-learning and online courses organizations and are accessible as online resources for the benefit of others. As mentioned in Table 1, some frameworks are only focused to quality in their respective region such as Europe [9], Australia [11], South Africa [12], Asia [17]. To maintain the necessary level of quality within their organization, e-learning leadership can design and implement online learning using these current frameworks based on their circumstances and requirements.

Therefore, based on the established frameworks and standards, we proposed our suggested design, which segregates the identified indicators into three major groups while taking into account all potential applications of e-learning. These are Academic, Technical, and Examination practices which serve as the major areas for our proposed Quality Teaching and Evaluation Framework. It is a condensed multidimensional framework used to evaluate teachers and e-learning courses or programs. The three components of the given

QTEF have included the assessment and evaluation component, technological aspects of the course delivery, design and development of the content, along with the indicators for management and policies of the course. Some of the key criteria of QTEF highlighted under each component are shown in Figure 2. The QTEF frameworks can be used to deliver courses at a myriad of study levels in an institution, school, or college.

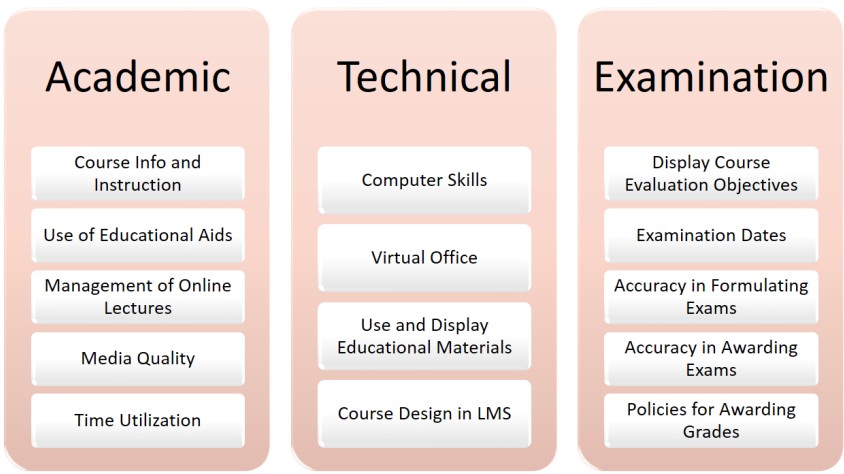

**Figure 2.** QTEF core components.

Our proposed Quality Teaching and Evaluation Framework is a multidimensional framework that consists of Academic, Technical, and Examination practices as a core domain for evaluating the teachers. The QTEF developed in Tables 2–4 produce outcomes that can be used to train and empower faculty, as well as provide guidance and improve course quality. There are 36 criteria, each with a weightage of 1–4 points. The overall evaluation points are set to 100, which are distributed in 40-40-20 on Academic (AC-1:AC-13), Technical (TC-1:TC15), and Examination (EC-1:EC-8) domains, respectively.

1. Academic Criteria. Academic criteria are crucial for e-learning evaluation since they guarantee that the given information is of exceptional quality and complies with industrial or educational standards. These criteria ensure that students obtain a comprehensive education that equips them for success in their chosen fields. The integration of the online course, quality of instructional design including clarity and organization of the course materials, qualifications of the instructors, and the resources provided for learners with established curricular standards is a critical academic requirement. It follows that the course material must be effectively incorporated into the broader educational program and should align with the learning objectives and results listed in the curriculum [25–27].

2. Technical Criteria. Technical criteria cover the technicalities of an e-learning course or program, including the needed hardware and software, the platform or method of distribution, and the course's accessibility for students with impairments. The learner's capacity to access and utilize the course materials may be impacted by these characteristics, making them crucial for evaluating e-learning. The technological aspects of a course can impact the overall user experience in e-learning. Technical problems with the course, such as numerous errors or lengthy loading times, can be annoying to the students and negatively affect their interest and retention. Overall, it is critical to consider technological factors when evaluating an e-learning program to ensure that all students can access and benefit from it and that the course materials are provided smoothly and seamlessly [28,29].

3. Examination Criteria. The purpose of examination criteria in e-learning evaluation is to ensure that evaluations are valid, reliable, and fair. In order to be fair, all students must be given an equal chance to show off their knowledge and abilities. The term "reliability" relates to the assessment's consistency and stability, which means it ought

to yield the same results every time it is used. The degree to which an assessment captures what it is meant to capture is called validity. When assessing e-learning resources, there are several essential assessment criteria to consider. These include relevance, objectivity, authenticity, reliability, validity, and fairness. To make sure that e-learning resources are credible and effective instruments for evaluating learners' knowledge and skills, it is crucial to carefully take into account these evaluation criteria while developing and reviewing e-learning products [30–32].

**Table 2.** QTEF—academic criteria.

| Category | Serial Nos. | Criteria Nos. | Criteria Description | Maximum Points |
|---|---|---|---|---|
| Academic | 1 | AC-1 | Course Information and Instructions | 4 |
| | 2 | AC-2 | Purpose and Structure of Course | 4 |
| | 3 | AC-3 | Course learning objectives | 3 |
| | 4 | AC-4 | Learner Evaluation Criteria | 2 |
| | 5 | AC-5 | Modern Educational Materials | 2 |
| | 6 | AC-6 | Variety of Educational Materials | 2 |
| | 7 | AC-7 | Use of educational and explanatory aids whenever necessary | 4 |
| | 8 | AC-8 | Managing Lectures and Student Interactions | 4 |
| | 9 | AC-9 | Clear, Understandable, Explainable Voice during Lectures | 3 |
| | 10 | AC-10 | Utilization of Lecture Time | 3 |
| | 11 | AC-11 | Extent of adherence to lecture times | 3 |
| | 12 | AC-12 | Plan for the Educational Activities of the Course | 3 |
| | 13 | AC-13 | The extent of commitment to implementing the educational activities of the course | 3 |

**Table 3.** QTEF—technical criteria.

| Category | Serial Nos. | Criteria Nos. | Criteria Description | Maximum Points |
|---|---|---|---|---|
| Technical | 1 | TC-1 | Display the scientific material clearly | 3 |
| | 2 | TC-2 | Links to the privacy policy and External tools required | 3 |
| | 3 | TC-3 | Computer skill | 3 |
| | 4 | TC-4 | Presentation of the Learning objectives in LMS | 3 |
| | 5 | TC-5 | Design and Easiness of the Course | 3 |
| | 6 | TC-6 | Course Organization | 3 |
| | 7 | TC-7 | Use of Interactive Tool/Material during Class | 2 |
| | 8 | TC-8 | Media Lectures | 3 |
| | 9 | TC-9 | Information Availability about tools use in course | 3 |
| | 10 | TC-10 | Educational Materials and Resources | 3 |
| | 11 | TC-11 | Display of Educational Material during Class | 2 |
| | 12 | TC-12 | Forums and Discussion | 3 |
| | 13 | TC-13 | Course Activities Advertisement/Announcement | 2 |
| | 14 | TC-14 | Virtual Office House | 2 |
| | 15 | TC-15 | Links about the University's services, etc. | 2 |

**Table 4.** QTEF—examination criteria.

| Category | Serial Nos. | Criteria Nos. | Criteria Description | Maximum Points |
|---|---|---|---|---|
| Examination | 1 | EC-1 | Evaluation measures specific learning objectives | 3 |
| | 2 | EC-2 | Clarity of the policy followed in distributing, monitoring and evaluating course grades | 2 |
| | 3 | EC-3 | Extent of adherence to exam dates | 2 |
| | 4 | EC-4 | Accuracy in formulating Assessments tasks | 3 |
| | 5 | EC-5 | Comprehensive Examination | 2 |
| | 6 | EC-6 | Coverage of learning outcomes in Assessment tasks | 3 |
| | 7 | EC-7 | Appropriate tests to measure learning outcomes | 3 |
| | 8 | EC-8 | Accuracy in Awarding grades | 2 |

## 4. Machine Learning and Performance Evaluation

### 4.1. Machine Learning

Machine learning is a rapidly growing field that has the potential to revolutionize various industries, including education. It is a valuable tool for educational policymakers and administrators, as it can help them allocate resources more effectively and make informed decisions about teacher retention and development. One area in which machine learning could be beneficial is in the prediction of teacher performance. By using machine learning algorithms to analyze data on teacher characteristics and behaviors, it may be possible to accurately predict which teachers are most likely to be effective in the classroom.

It delivers automated e-learning course delivery through cutting-edge LMS systems, intelligent algorithms, and online learners of the future [33]. A branch of computing algorithms called machine learning is constantly developing and aims to replicate human intelligence by learning from the environment. In the brand-new era of "big data", they are regarded as the workhorse [34]. Machine learning techniques can be applied in various ways to enhance e-learning platforms. The machine learning approach is used in this study to build a high-quality framework model for forecasting instructor performance and identifying potential strengths and weaknesses of the specific faculty member.

The crux of this study is developing a framework model that allows the assessment criteria to be identified and categorized to determine which criteria are more helpful in assessing teacher performance using machine learning. Depending on their posts in the discussion boards, in-person facial expressions, or other methods that can assist teachers in identifying students who need more attention and motivation, ML techniques can play a significant role in identifying frustrated or dissatisfied learners [35]. Additionally, e-learning can be automated, and decisions about updating the activities and materials utilized for learning can be made using machine learning techniques [36,37].

### 4.2. Machine Learning Algorithm

Machine learning algorithms are classified into ontologies based on the intended result of the algorithm. ML types can be categorized into supervised, unsupervised, semi-supervised, and reinforcement learning [38]. Although, there are many different types of machine learning algorithms, the choice of which one to use will depend on the specific goals of the study and the nature of the data.

Some commonly used algorithms for prediction tasks include linear regression, decision trees, support vector machines, and random forests. One of the supervised learning-type algorithms is Linear Regression, which executes a regression operation and is commonly used for predictive analysis. Two theories are approached via regression. First, regression analyses are frequently used for forecasting and prediction, areas in which machine learning and their application have a lot in common. Second, in some circumstances, causal relationships between the independent and dependent variables can be ascertained using regression analysis.

Regression uses independent variables to model a goal prediction value. It is mainly used to determine how variables and predictions relate to one another. It is significant to highlight that regressions alone can only show correlations between a dependent variable and a fixed dataset collection of other factors [39]. Regression can be classified into two broad categories based on the number of independent variables.

- Simple Linear Regression (LR): A linear regression procedure is referred to as simple linear regression if only one independent variable is utilized to predict the outcome of a numerical dependent variable.
- Multiple Linear Regression (MLR): A linear regression process is referred to as multiple linear regression if it uses more than one independent variable to anticipate the value of a numerical dependent variable.

## 5. Methodology

### 5.1. Dataset and Data Description

A dataset is a grouping of different kinds of data that have been digitally preserved. Any project using machine learning needs data as its primary input and leaves a significant determinant of the performance of learning models. The dataset represents the real-world problem and is properly pre-processed to remove any biases or noise [40,41]. We first need to gather a dataset containing relevant data on teacher characteristics and behaviors to predict teacher performance using machine learning. It could include data on a teacher's teaching style, use of instructional strategies, engagement with students, and examination conduct.

Figure 3 presents the layout of the dataset which we have used for training and testing our ML model. The dataset contains 400 rows of records and 41 columns containing the 13 Academic (AC), 15 Technical (TC), and 08 Examination (EC) criteria as presented in Tables 2–4. The dataset contains performance evaluations of 219 male and 181 female members from different university departments. The data collection was done through a performance evaluation dataset that was generated from the faculty members of Jazan University who were teaching distance learning courses for undergraduate students in English, Arabic, and Journalism programs. First, the participating faculty members completed training on how to use the LMS, as well as being fully briefed on the framework requirements and grading scale on which the quality framework was designed. Later, on the completion of the course, the Quality unit of the university evaluates the e-course and awards grades based on the defined QTEF standards.

| | Gender | AC-1 | AC-2 | AC-3 | AC-4 | AC-5 | AC-6 | AC-7 | AC-8 | AC-9 | ... | EC-1 | EC-2 | EC-3 | EC-4 | EC-5 | EC-6 | EC-7 | EC-8 | Examination | Final |
|---|---|---|---|---|---|---|---|---|---|---|---|---|---|---|---|---|---|---|---|---|---|
| 0 | Male | 3.0 | 3.0 | 3 | 2 | 3 | 2 | 2 | 3.0 | 2.0 | ... | 2.0 | 2 | 2 | 2 | 2 | 1 | 3 | 3 | 17.0 | 91.0 |
| 1 | Female | 1.5 | 2.0 | 3 | 0 | 3 | 2 | 2 | 3.0 | 2.0 | ... | 2.0 | 2 | 2 | 2 | 2 | 1 | 3 | 0 | 14.0 | 66.0 |
| 2 | Female | 1.5 | 2.0 | 0 | 0 | 0 | 0 | 0 | 2.5 | 2.0 | ... | 2.0 | 0 | 0 | 0 | 0 | 1 | 0 | 0 | 3.0 | 43.0 |
| 3 | Male | 3.0 | 3.0 | 3 | 2 | 3 | 2 | 2 | 3.0 | 2.0 | ... | 2.0 | 2 | 2 | 2 | 2 | 1 | 3 | 3 | 17.0 | 91.0 |
| 4 | Female | 1.5 | 2.0 | 3 | 0 | 3 | 2 | 2 | 3.0 | 2.0 | ... | 2.0 | 2 | 2 | 2 | 2 | 1 | 3 | 0 | 14.0 | 65.9 |
| 5 | Female | 1.5 | 2.0 | 3 | 0 | 3 | 2 | 2 | 3.0 | 2.0 | ... | 2.0 | 2 | 2 | 2 | 2 | 1 | 3 | 0 | 14.0 | 69.5 |
| 6 | Male | 3.0 | 3.0 | 3 | 2 | 3 | 2 | 2 | 3.0 | 2.0 | ... | 2.0 | 2 | 2 | 2 | 2 | 1 | 3 | 3 | 17.0 | 82.6 |
| 7 | Male | 3.0 | 3.0 | 3 | 2 | 3 | 2 | 2 | 3.0 | 1.7 | ... | 1.0 | 2 | 2 | 2 | 2 | 1 | 3 | 3 | 16.0 | 82.6 |
| 8 | Male | 3.0 | 3.0 | 3 | 2 | 3 | 2 | 2 | 3.0 | 2.0 | ... | 2.0 | 2 | 2 | 2 | 2 | 1 | 3 | 3 | 17.0 | 90.1 |
| 9 | Male | 3.0 | 3.0 | 3 | 2 | 3 | 2 | 2 | 3.0 | 2.0 | ... | 2.0 | 2 | 2 | 2 | 2 | 1 | 3 | 3 | 17.0 | 80.2 |

**Figure 3.** Dataset for ML model.

The data were collected through the well-defined mechanism for evaluating course profiles on Blackboard—Learning Management System and the other associated factors defined in QTEF. The dataset Summary Data from Table 5 contains each score's minimum, maximum, and specific percentiles. The standard mean of the final score from the dataset was calculated as 67.26, while its standard deviation was 17.52. Table 5 shows that 50% of the teachers receive more points than the norm and that 75% have more than 84 points. Furthermore, Linear Model Plot (lmplot) is a function used in the Python library seaborn [42] that allows us to create a scatterplot with a linear regression line. The lmplot is often used to visualize the relationship between two variables in a dataset in machine learning research, it combines the regplot () and Facet Grid. When the data are displayed as an lmplot and analyzed, the independent variable (Academic, Technical, and Examination criteria score) combines regplot() and Facet Grid, making the Final column the dependent variable. When combined, they offer a simple method of seeing the regression line in a faceted graph. Figure 4 presents the gender-wise scatter plots with overlaid regression lines and shows the relationship between the QTEF categories and the impact on the final score of the teacher performance matrix. They also demonstrate the statistical technique that

relates a dependent variable (Final Grade) to independent variables (Academic Criteria, Technical Criteria, Examination Criteria score).

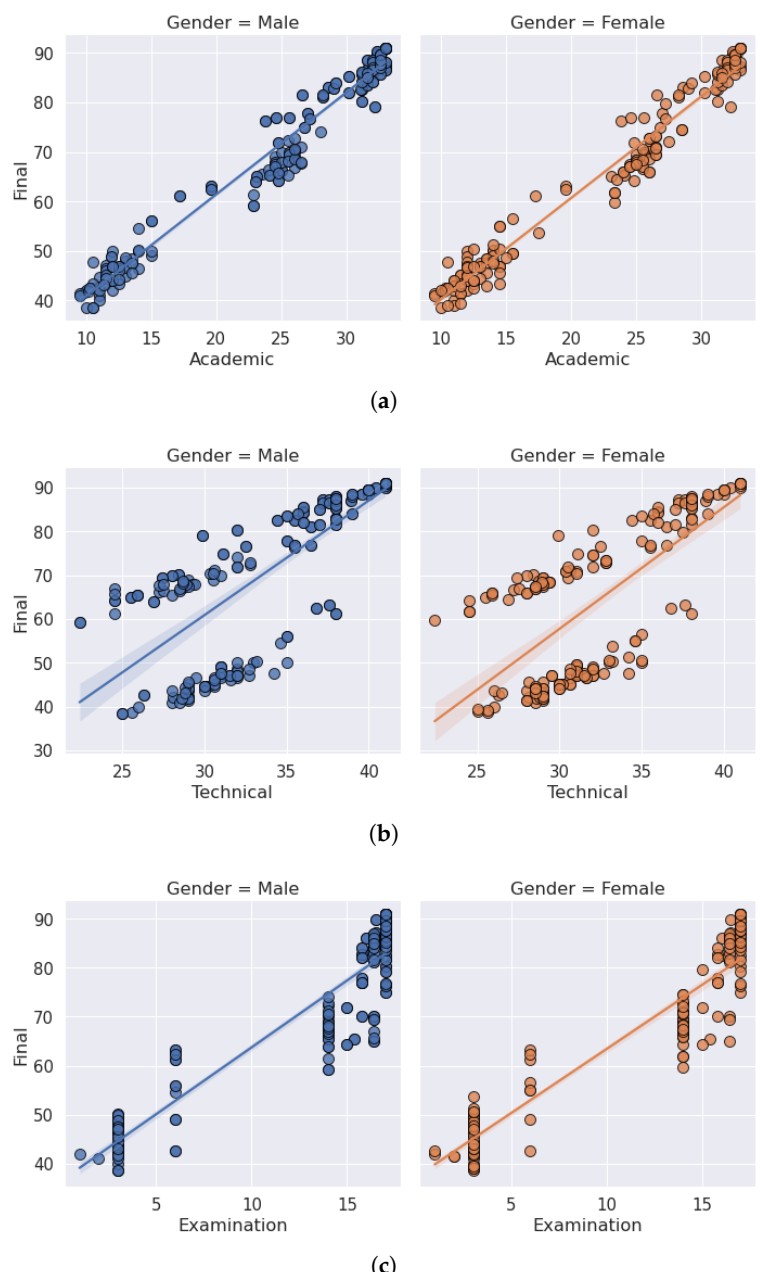

**Figure 4.** Final grades distribution—gender wise. (**a**) Academic criteria, (**b**) Technical criteria, (**c**) Examination criteria.

**Table 5.** Dataset description.

| Sr. No | Parameter | Final Score Value |
|--------|-----------|-------------------|
| 1 | Records Count | 400 |
| 2 | Mean | 67.26 |
| 3 | Standard Deviation | 17.52 |
| 4 | Minimum Score | 38.50 |
| 5 | 25% | 47.57 |
| 6 | 50% | 68.70 |
| 7 | 75% | 84.50 |
| 8 | Maximum Score | 91.00 |

*5.2. Correlation*

In machine learning, correlation refers to the relationship between two variables and how they change concerning each other [43]. A high correlation between two variables means that they are strongly related and that a change in one variable is likely to be accompanied by another variable. Conversely, a low correlation between two variables means that they are not strongly related, and a change in one variable is unlikely to be accompanied by a change in the other variable. When building a Machine learning model, understanding a dataset is crucial, and heatmaps are just one of the numerous tools at a data scientist's command. When used appropriately, the correlation matrix in the heatmap style is an extremely powerful tool to identify the low/high correlated variables.

In order to simplify the feature selection process, the user will be able to discover highly associated variables. They are used to finding out the dependent and independent variables for the learning model. Table 6 lists the criteria that were selected for the development of the multiple linear regression model. In our defined framework, out of the 36 presented criteria, three factors from each evaluation category were selected as independent variables to have a high/low impact on the dependent variable. Figure 5 presents the high and low correlation between all criteria and the final score. Each indicator of our proposed framework (variable) is represented on both axes of the graph, and color is used to indicate how each feature's relationship to other variables changes. It is clear that the variables are more strongly connected as the color gets darker in either direction.

**Table 6.** Selected QTEF criteria.

| Criteria Category | Criteria Number | Correlation Type |
|---|---|---|
| Academic | AC-6, AC-7, AC-12 | High |
| Technical | TC-4, TC-9, TC-15 | High |
| Examination | EC-4, EC-5, EC-7 | High |
| Academic | AC-9, AC-10, AC-13 | Low |
| Technical | TC-2, TC-3, TC-13 | Low |
| Examination | EC-1, EC-6, EC-8 | Low |

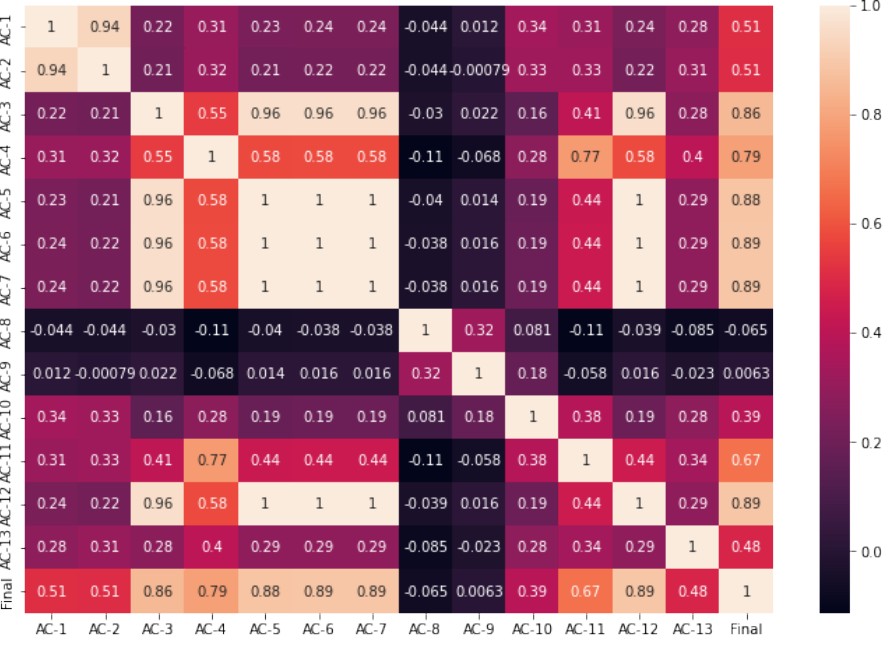

(a)

**Figure 5.** *Cont.*

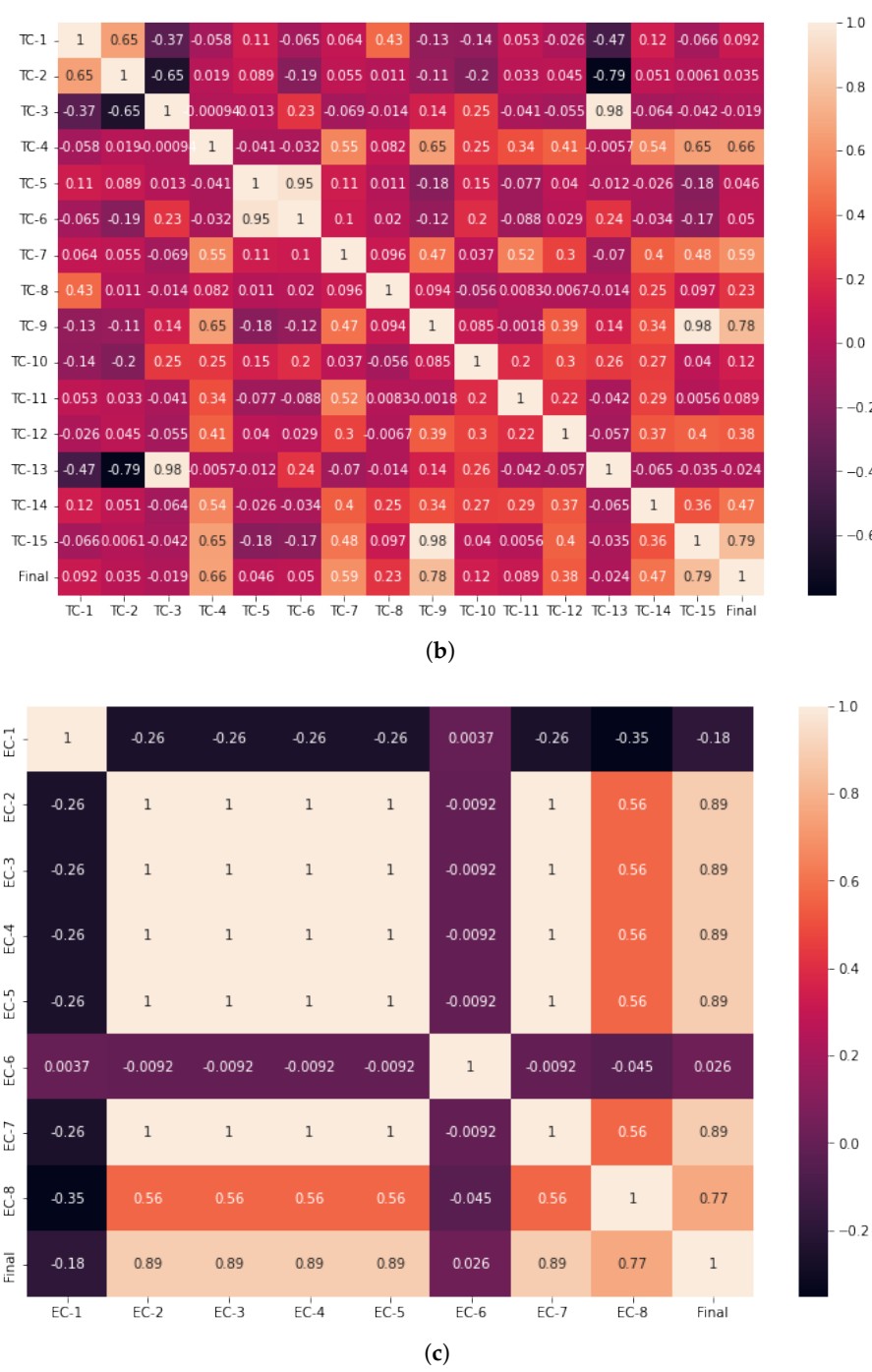

**Figure 5.** Correlation graphs—dependent and independent variable selection. (**a**) Academic criteria correlation. (**b**) Technical criteria correlation. (**c**) Examination criteria correlation.

## 6. Results and Discussion

This section presents the testing results for the machine learning algorithm used on the dataset. As mentioned in the previous section, the dataset consists of teacher evaluation records based on their academic, technical, and examination criteria. The popular machine-learning python libraries pandas [44] for data analysis and statistics, as well as Scikit-learn [45] to train and test the models are used in Python for the ML mode. All models were executed using the Google Colaboratory (Colab) machine learning tool. Google Colaboratory is particularly useful for data science and machine learning applications, as it provides access to powerful computing resources and libraries. One of the critical benefits of Colab is that it allows that we can run code in the cloud, meaning

we do not need to have a powerful computer or install any software on a local machine. Furthermore, using Google Colab notebooks in the teaching and learning process can help students become familiar with programming ideas [46].

To calculate performance, multiple regression analysis is done on the QTEF data presented in Tables 2–4 in the ratio of 65:35, i.e., training (260 records) and testing (140 records) dataset. Keeping view of the least number of dataset records, the goodness of fit is determined by the coefficient of determination (R2). It is a statistic applied to statistical models whose main objective is to either predict future outcomes or test hypotheses using data from other relevant sources. Based on the percentage of the overall variation in outcomes that the model is responsible for explaining, it provides a gauge of how effectively observed results are mirrored by the model [40,41]. R2 can be anywhere between 0.0 and 1.0. If the determination coefficient is 0, the value of the dependent variable, Y, cannot be predicted from the value of the independent variable, X. As a result, it can be concluded that there is no linear relationship between the variables, and a horizontal line provides the best fit. The median of all Y values is where the line crosses. All points fall perfectly on the horizontal path without scattering if their value is 1.0. An exact prediction of Y can be made if X is known.

The R2 value for the multiple regression results in Table 7 based on highly correlated variables from Figure 2 is achieved as 0.9322, which shows that predictor variables have 93.22% accuracy. Causation is not necessarily implied by correlation. Although a significant connection would suggest causation, there may be alternative reasons. The factors may appear related, but there may be no real underlying relationship. Therefore, it could be the outcome of random chance [47]. As a result, we have replaced TC-14 with TC-15 and EC-5 to EC-3 as Scenario-1, which increases the accuracy to 95.35%. In addition, we have found that the replacement of examination criteria to EC-1 from EC-3 and technical from TC-4 to TC-7 in the training model increases the best possible accuracy of 95.77%. Additionally, the Mean Square Error has decreased from 20.12 to 13.78 and finally reduced to 12.56, measuring the squared average difference between the actual and anticipated data.

**Table 7.** Results and accuracy.

| Correlation Type | Selected Criteria | R2 Value | Mean Square Error |
|---|---|---|---|
| Highly Correlated | AC-6, AC-7, AC-12, TC-4, TC-9, TC-15, EC-4, EC-5, EC-7 | 0.9322 | 20.12 |
| Adjusted Variable—Scenario (1) | AC-6, AC-7, AC-12, TC-4, TC-9, TC-14, EC-3, EC-4, EC-7 | 0.9535 | 13.78 |
| Adjusted Variable—Scenario (2) | AC-6, AC-7, AC-12, TC-7, TC-9, TC-14, EC-1, EC-4, EC-7 | 0.9577 | 12.56 |

The findings from the testing vs. training dataset of the machine learning model using Python code are represented as Figure 6 and graphically as Figure 7. The results demonstrate that the projected and actual figures are sufficiently close to one another by correlating them. Our MLR model produces results nearly identical to the distribution of the actual outcomes, as was anticipated by the grades distribution given in Table 5. It is because of the best-selected indicators from our QTEF model. Nevertheless, a few exceptions may be found in Figure 6 due to the use of limited dataset records.

|     | Actual Value | Predicted Value | Difference |
| --- | --- | --- | --- |
| **0** | 71.1 | 71.163939 | -0.063939 |
| **1** | 87.0 | 86.983114 | 0.016886 |
| **2** | 69.5 | 72.422604 | -2.922604 |
| **3** | 88.6 | 86.983114 | 1.616886 |
| **4** | 48.0 | 48.969098 | -0.969098 |
| **...** | ... | ... | ... |
| **135** | 46.1 | 48.969098 | -2.869098 |
| **136** | 86.6 | 85.724448 | 0.875552 |
| **137** | 76.9 | 83.262803 | -6.362803 |
| **138** | 64.3 | 60.650495 | 3.649505 |
| **139** | 81.6 | 86.983114 | -5.383114 |

140 rows × 3 columns

**Figure 6.** Results variations—Scenario (2).

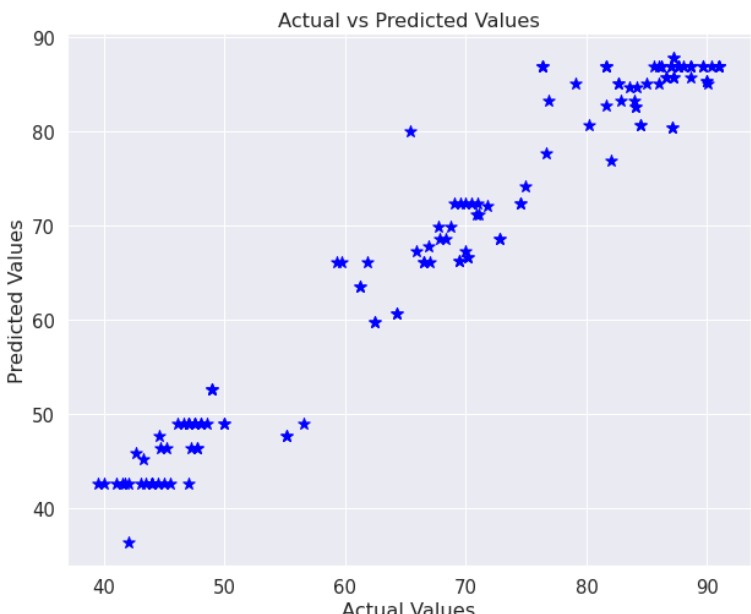

**Figure 7.** Adjusted variable results—Scenario (2).

## 7. Conclusions

In conclusion, using machine learning approaches to assess teacher effectiveness has the potential to yield insightful information and promote the ongoing development of instructional strategies. Machine learning algorithms can discover patterns and correlations that may take time to be evident to people by evaluating vast amounts of data, such as instructor observations made during the delivery of courses. Academic administrators can use this to pinpoint areas of strength and weakness in their teaching faculty and best practices that can be applied throughout the institution.

This paper uses the machine learning technique on a teacher quality matrix to predict teacher performance on a particular course based on the defined Quality Teaching and Evaluation Framework. As many machine learning approaches are used for data classification, the regression method is used here. Information such as academic, technical, and

examination criteria is defined and marked to predict the teacher's performance at the end of the semester. This study will help teachers improve their teaching performance, ultimately enhancing student performance. Similarly, it will also work to identify those teachers who need special attention to better perform in particular criteria and take appropriate action for their teaching during the semester. There are two potential limitations in this study that could be addressed in future research. First, the study focused only on a limited sample of 400 datasets. Second, it only uses multiple regression for machine learning. Furthermore, we believe that the accuracy of the study's findings can be increased by including other evaluation criteria, such as teacher qualifications, age, experience, and nationality. By contrasting the MLR model with other ML algorithms, such as decision trees, random forests, SVM, and others, the accuracy of the results may also be confirmed.

Applying machine learning to teacher performance reviews offers a fascinating chance to deepen our understanding of suitable teaching methods and promote educational progress. It is crucial to remember, though, that machine learning is not a fix-all for all problems relating to teacher effectiveness. The necessity for high-quality data inputs, the possibility of algorithmic bias, and the requirement for continuing maintenance and changes to the machine-learning model are a few potential limits to consider.

**Author Contributions:** Conceptualization, A.A., K.M.N. and M.N.S.; methodology, K.M.N. and M.N.S.; software, M.N.S.; validation, K.M.N. and A.A.; formal analysis, K.M.N. and M.N.S.; investigation, K.M.N. and A.A.; resources, A.A.; data curation, K.M.N. and M.N.S.; writing—original draft preparation, A.A., K.M.N. and M.N.S.; visualization, M.N.S.; supervision, A.A.; project administration, A.A. All authors have read and agreed to the published version of the manuscript.

**Funding:** This research received no external funding.

**Institutional Review Board Statement:** Not applicable.

**Informed Consent Statement:** Informed consent was obtained from all subjects involved in the study.

**Data Availability Statement:** The data presented in this study are available on request from the corresponding author.

**Conflicts of Interest:** The authors declare no conflict of interest.

## Abbreviations

The following abbreviations are used in this manuscript:

| | |
|---|---|
| QTEF | Quality Teaching Evaluation Framework |
| ML | Machine Learning |
| Colab | Google Colaboratory |
| MLR | Multiple Linear Regression |
| LR | Simple Linear Regression |

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
