# Peer review of "Academic Teaching Quality Framework and Performance Evaluation Using Machine Learning"

_applsci, doi:10.3390/app13053121_

Round 1
Reviewer 1 Report
Thank you for submitting your article “Academic Teaching Quality Framework and Performance Evaluation using Machine Learning” for review and consideration for publication in the Applied Sciences journal.
The present paper proposed a Quality Teaching and Evaluation Framework and validated it using a machine learning approach. I agree with the authors that higher education institutions should continuously strive in giving their learners a high-quality education. The literature is also in agreement, which is why there are several existing frameworks similar to this proposal. Some examples include:
Shawn R. Simonson, Brittnee Earl & Megan Frary (2022) Establishing a Framework for Assessing Teaching Effectiveness, College Teaching, 70:2, 164-180, DOI: 10.1080/87567555.2021.1909528
Satsope, M. R., John, M. K., Kabelo, C., & Mahlapahlapana, T. (2015). Towards a Framework for Evaluating Quality Teaching in Higher Education. Mediterranean Journal of Social Sciences, 6(4), 223. Retrieved from https://www.richtmann.org/journal/index.php/mjss/article/view/7000
Despite the presence of existing frameworks, the present paper failed to acknowledge the literature. Doing so can make it appear as though the authors are not familiar with the existing body of knowledge on the topic, which can undermine the credibility of this paper. Thus, they must thoroughly examine these frameworks, compare their strengths and weaknesses, and identify the gaps that could open an opportunity to conduct another study and propose a new or updated framework.
In their proposed framework, the authors used Academic, Technical, and Examination practices as the primary components. It would be interesting to add a diagram that visualize the relationships (if there are) between these components. If there are indeed connections between them, the authors must revise the literature review and emphasize how one affects the other.
As the framework is limited to these three criteria, the authors may also explain why other predicting factors of quality online education were not included. One example is organizational factors that have been linked to e-learning system quality (See Al Mulhem, 2022; DOI: 10.1080/2331186X.2020.1787004).
Another important factors that may be considered is the sociodemographic profiles of students. According to Garcia (2022; DOI: 10.4018/978-1-6684-4364-4), there are inequities and inequalities in accessing online educational opportunities. Although this framework aims to measure the quality performance of teachers, the authors may offer some discussion on how students' profile may have influenced the quality of the online education that is beyond the scope of the framework. Given the digital divide, students may not be able to enjoy the quality performance of teachers if they do not have access to these opportunities in the first place.
The dataset presented in Table 4 does not offer any clear and detailed description of the dataset. What are the attributes (columns) available in this dataset? The authors may also provide samples to help readers understand the data better.
How many data from this dataset was used for testing and for training? The dataset is small the authors should discuss whether it is sufficient and the potential implications or consequences of having a small dataset.
In the same light, results presented in Figures 1 and 2 need elaboration. What are these findings and what are their implications?
The authors concluded that they used a machine learning approach to predict teacher performance on a particular course based on their framework. An elaboration on how prediction was or can be made should be present in the study.
Although they are not always required, the authors may include the study limitations
Author Response
- Existing frameworks
- The existing framework/benchwork/indicators along with their key domain parameters added in Table 1. Quality Benchmarks-Frameworks for E-Learning.
- Framework Visualization
- All criteria presented in QTeF have their own indication in a particular domain, however, Figure 1. Quality Indicators for Digital Education added shows the general framework/benchmark offers.
- All criteria presented in QTeF have their own indication in a particular domain, however, Figure 1. Quality Indicators for Digital Education added shows the general framework/benchmark offers.
- Limited to three criteria
- The QTeF is based on the previous work, however, it is a condensed version with a segregation of domains into Academic, Technical and Examination exclusively for e-learning courses.
- The QTeF is based on the previous work, however, it is a condensed version with a segregation of domains into Academic, Technical and Examination exclusively for e-learning courses.
- Dataset
- The dataset, collected from Jazan University, has 400 records and is used as an ML model in 65-35 ratio. The exact number of records, parameters, python generated graph, and tables are presented in Figures 2 and 5.
- Limitation and Future Work
- The limitation of the proposed QTeF was added along with the future work to enhance the ML model by comparing it with different ML algorithms presented in the Conclusion section.
Reviewer 2 Report
This paper is interesting from the scientific point of view and raises very current topics of e-learning supported by modern technologies. The paper layout is correct.
However, authors should rethink/improve:
- In Abstract - major findings as a result of your analysis; and, a brief summary of your interpretations and conclusions should be added;
- In Introduction - beyond the preparation of the background and brief discussion of the basic references related to the topic, authors should: indicate a research gap; announce the realized research, clearly indicating what is novel and why it is significant, and present the main purpose of paper.
- In Literature review - there are many pieces of text that are clearly prepared from earlier research papers, but references are missing. The number of references in the text should be increased (completed), and the authors should use very up-to-date sources - the discussed topic is characterized by high dynamism of changes.
- In Methods – data collection is very poorly described. There is no information when data was collected, why in this way and scope, and such a phrase "our university" should not appear in the paper.
- In Results and Discussion – if authors want to provide the discussion, they should provide a confrontation of the achieved results with previously published papers, their opinion of established differences, and their attitude to the results.
- In Conclusion – authors should add the explanation how the research has created knowledge or at least moved the body of knowledge forward. Moreover, advantages, limitations, possible applications, and future research direction should be presented or described in much more detail.
- There are some spelling and grammar mistakes, which should be improved.
Author Response
- Abstract
- The major finding of the Results is the Technical indicators' variation added in the Abstract along with further elaboration in different sections of the Paper.
- The major finding of the Results is the Technical indicators' variation added in the Abstract along with further elaboration in different sections of the Paper.
- Introduction
-
The existing framework/benchwork/indicators along with their key domain parameters added in Table 1. Quality Benchmarks-Frameworks for E-Learning.
-
All criteria presented in QTeF have their own indication in a particular domain, however, Figure 1. Quality Indicators for Digital Education added shows the general framework/benchmark offers.
-
- Literature Review
- Added relevant references and go through the Turnitin results.
- Methods
- The dataset, collected from Jazan University, has 400 records and is used as an ML model in 65-35 ratio. The exact number of records, parameters, python generated graph, and tables are presented in Figures 2 and 5.
- Results
- The results graph and tables, particularly for the ML model, are explained and updated.
- The results graph and tables, particularly for the ML model, are explained and updated.
- Conclusion
- This section updated along with the limitation of the proposed QTeF was added along with the future work to enhance the ML model by comparing it with different ML algorithms presented in the Conclusion section.
Reviewer 3 Report
The paper is very poorly written:
1- Abstract should be more specific
2- Introduction must be clearly present the background, purpose of the study, scope of the study, and summarization of the results;
3- The literature review not exist
4- The methodology and results need to be more clearly and explained in the discussion with the comparison with others
Author Response
- Abstract
- The major finding of the Results is the Technical indicators' variation added in the Abstract along with further elaboration in different sections of the Paper.
- The major finding of the Results is the Technical indicators' variation added in the Abstract along with further elaboration in different sections of the Paper.
- Introduction
-
The existing framework/benchwork/indicators along with their key domain parameters added in Table 1. Quality Benchmarks-Frameworks for E-Learning.
-
All criteria presented in QTeF have their own indication in a particular domain, however, Figure 1. Quality Indicators for Digital Education added shows the general framework/benchmark offers.
-
Added relevant references and go through the Turnitin results.
-
- Methods
- The dataset, collected from Jazan University, has 400 records and is used as an ML model in 65-35 ratio. The exact number of records, parameters, python generated graph, and tables are presented in Figures 2 and 5.
- Results
- The results graph and tables, particularly for the ML model, are explained and updated.
- The results graph and tables, particularly for the ML model, are explained and updated.
- Conclusion
- This section updated along with the limitation of the proposed QTeF was added along with the future work to enhance the ML model by comparing it with different ML algorithms presented in the Conclusion section.
Round 2
Reviewer 1 Report
I would like to start by saying that I am impressed with the revisions made to the paper. The authors have done an excellent job in clarifying some of my feedback and providing more detailed explanations in the manuscript. The addition of new figures and tables also helps to better illustrate the findings.
In particular, I appreciated the efforts made to address my previous comment on the existing literature, especially by adding the Table 1. While this new information paints the current state of the frameworks, it is unclear how these frameworks led to the development of the authors' proposed framework. I would like to reiterate my previous comment:
"...they must thoroughly examine these frameworks, compare their strengths and weaknesses, and identify the gaps that could open an opportunity to conduct another study and propose a new or updated framework."
Rather than the Quality Indicators for Digital Education in Figure 1, it should have been the visual representation of the QTeF as indicated in my previous comment. It is also as important to emphasize the connection between Academic, Technical, and Examination practices.
In addition to these partial compliance, I believe the authors failed to address the following from my previous comments:
1. Justifications on why other predicting factors of quality online education were not included.
2. Role of sociodemographic profile of students.
Other than these, I believe that the revisions have significantly improved the quality of the paper and it is now much stronger and more complete.
Author Response
- Table.1 Modification
- The modified Table 1 added, which also shows the parameters of each framework/benchmark.
- Previous Framework vs QTEF / Other Predicting Factors
-
The frameworks/benchmarks shown in the Table 1 have been developed by leading e-learning and online courses organizations and are accessible as online resources for the benefit of others.
-
As mentioned in Table 1, some frameworks are only focused to respective region like Europe[37], Australia[39], South Africa[40], Asia[45].
-
To maintain the necessary level of quality within their organization, e-learning leadership can design and implement online learning using these current frameworks based on their circumstances and requirements.
-
Therefore, based on the established frameworks and standards, we proposed our suggested design, which segregates the identified indicators into three major groups while taking into account all potential applications of e-learning.
-
- Student Roles
- The QTEF is mainly focused on delivering lectures, therefore no factor for student evaluation added in the criteria list.
- The same presented criteria evaluated by Machine Learning.

Reviewer 3 Report
Can be accepted
Author Response
Thanks.